# INVERSE ATTENTION AGENTS FOR MULTI-AGENT SYSTEMS

**Qian Long**[*]
UCLA
longqian@ucla.edu

**Ruoyan Li**[*]
UCLA
liruoyan2002@ucla.edu

**Minglu Zhao**[*]
UCLA
mz517@ucla.edu

**Tao Gao**
UCLA
tao.gao@stat.ucla.edu

**Demetri Terzopoulos**
UCLA
dt@cs.ucla.edu

## ABSTRACT

A major challenge for Multi-Agent Systems is enabling agents to adapt dynamically to diverse environments in which opponents and teammates may continually change. Agents trained using conventional methods tend to excel only within the confines of their training cohorts; their performance drops significantly when confronting unfamiliar agents. To address this shortcoming, we introduce Inverse Attention Agents that adopt concepts from the Theory of Mind (ToM) implemented algorithmically using an attention mechanism trained in an end-to-end manner. Crucial to determining the final actions of these agents, the weights in their attention model explicitly represent attention to different goals. We furthermore propose an inverse attention network that deduces the ToM of agents based on observations and prior actions. The network infers the attentional states of other agents, thereby refining the attention weights to adjust the agent's final action. We conduct experiments in a continuous environment, tackling demanding tasks encompassing cooperation, competition, and a blend of both. They demonstrate that the inverse attention network successfully infers the attention of other agents, and that this information improves agent performance. Additional human experiments show that, compared to baseline agent models, our inverse attention agents exhibit superior cooperation with humans and better emulate human behaviors.

## 1 INTRODUCTION

Multi-Agent Reinforcement Learning (MARL) has significantly advanced the study of complex, interactive behaviors in multi-agent systems, allowing intricate modeling of scenarios involving multiple autonomous agents. However, creating an ad-hoc agent that excels with various types of teammates and opponents poses a significant challenge. Current methods suffer a limitation: While agents trained together demonstrate proficient coordination, their performance deteriorates markedly when collaborating with unfamiliar agents.

To address this challenge, we delve deeper into multi-agent collaboration by incorporating a cognitive perspective, specifically through the modeling of attention and Theory of Mind (ToM) (Bratman, 1987) within the MARL framework. Unlike classical ToM research, which focuses on attributing mental states such as beliefs and desires, our model shifts to the crucial yet less-emphasized component of *attention*. Our methodology adopts a mentalistic approach, explicitly modeling the internal attentional state of agents using an attention recognition neural network that can be trained end-to-end in combination with components of MARL. Contrary to traditional ToM modeling approaches that rely heavily on Bayesian inference to handle mental state transitions (Baker et al., 2009; Kleiman-Weiner et al., 2016; Shum et al., 2019; Gao et al., 2020; Tang et al., 2020), our method maintains the ontology of these states while focusing on the direct modeling of agents' attentional mechanisms. This shift from Bayesian methods to more direct, attention-based modeling opens new pathways for understanding and enhancing agent interactions in complex environments.

---

[*]Equal contribution

To enhance task performance, we craft an agent that adeptly infers the attention of other agents — a crucial aspect of ToM. We employ gradient field functions to construct goals within the environment. Subsequently, we incorporate a self-attention architecture within the policy network to generate attention weights for these goals, guiding the agent's actions accordingly. Throughout the training phase, we accumulate the weighted goals and corresponding actions into a training dataset. This dataset then serves to train an attention inference network that determines attention weights based on the observations and actions of other agents, effectively modeling their attention. In the final phase, we amalgamate the self-attention structure with the inverse attention to create our *Inverse-Attention Agent*. Our agent integrates environmental observations with inferred attention weights from the inference network to fine-tune its final actions.

We demonstrate the effectiveness of our approach through a series of experiments adapted from the Multi-agent Particles Environment (MPE) (Lowe et al., 2017). Specifically, we employ a Mix-and-Match scheme to evaluate the performance of our models in ad-hoc collaboration, by pairing models trained using various algorithms. The results indicate that our model consistently outperforms baselines, particularly excelling in cooperative tasks. Moreover, we conduct a series of human experiments where humans join teams alongside agents trained with different methods. Through both quantitative and qualitative analysis, our model demonstrates superior cooperation within ad-hoc teams comprising both humans and previously unseen agents.

To summarize, we introduce a MARL training scheme inspired by theories in cognitive science, where each agent explicitly reasons about the attentional states of group members. Our approach is specifically designed to enhance ad-hoc coordination among agents, addressing a long-standing challenge in MARL. Our framework also adopts a simplified state representation using social gradient fields, which further reduces the complexity of the environmental input that agents must process, thereby facilitating more flexible and efficient decision making. By explicitly modeling attention and incorporating cognitive principles, our approach paves the way for more effective multi-agent collaboration in complex, interactive scenarios.

## 2 RELATED WORK

### 2.1 THEORY OF MIND AND ATTENTION

Theory of Mind (ToM) refers to the cognitive ability to attribute mental states to oneself and others (Bratman, 1987). It allows for tailored strategies to be generated in multi-agent scenarios, where one can reason about what other players are doing and determine one's action accordingly. ToM has been an active area of research in multi-agent systems, with the goal of designing agents that coordinate more like humans. Previous work has primarily utilized Bayesian approaches to explicitly model beliefs, desires, and intentions (Shum et al., 2019; Kleiman-Weiner et al., 2016; Wu et al., 2021), providing a principled framework for inferring and updating beliefs about other agents' mental states based on observed behavior and actions. To avoid the complexity of recurrently reasoning about each others' mental state, Tang et al. (2020) and Stacy et al. (2021) proposed frameworks based on the idea of shared agency, relying on coordinating group-level mental states and establishing a shared understanding of the task and goals among agents. Departing from Bayesian methods, Rabinowitz et al. (2018) proposed to integrate ToM reasoning directly into the neural network architecture, aiming to learn representations of other agents' mental states in a data-driven manner.

Building upon previous frameworks, our work addresses three key aspects: First, we develop our model around the idea that ToM is not merely concerned with beliefs, desires, and intentions, but is a human-unique ability to be sensitive to what others are attending to. While attention as a critical mental state is often neglected in ToM research, our work aims to address this gap by explicitly modeling attention mechanisms. Second, we model cooperation from an individualistic perspective to encourage more flexible behavior generation. In this way, instead of shared agency, our agents maintain an individualistic perspective while coordinating with teammates. Third, we deviate from traditional Bayesian approaches to modeling ToM and instead develop an end-to-end neural network-based training, thus allowing for more flexibility and generalizability.

## 2.2 Ad-Hoc Teaming

Ad-hoc collaboration is defined as the challenge of enabling autonomous agents to effectively coordinate with previously unknown teammates, without any prior opportunities for coordination or agreement on strategies (Stone et al., 2010). Addressing this problem necessitates agents to model the behavior of their teammates and subsequently select actions that facilitate effective collaboration, while simultaneously adapting to changes or new information that emerges during the interaction. A prominent approach to modeling teammate behavior is type-based reasoning, which represents teammates as belonging to hypothesized behavior types (Barrett et al., 2017). Alternatively, neural network-based techniques have been proposed to infer teammate types from observations (Rabinowitz et al., 2018; Rahman et al., 2021; Xie et al., 2021). Once teammate models are obtained, agents perform downstream action planning with techniques such as Monte Carlo tree search (Barrett et al., 2014; Albrecht and Stone, 2019) and meta-learning action selection (Zintgraf et al., 2021). Adapting the agent's behavior based on new information about teammates is also crucial in sustaining the collaboration. Addressing this, Macke et al. (2021) employed communication between agents and Lupu et al. (2021) proposed a method to generate diverse training trajectories to improve generalization to novel teammates.

In our work, we tackle the ad-hoc problem by leveraging the attention mechanism. We argue that the instability issues in ad-hoc settings typically arise because agents fail to comprehend unseen states, consequently making it difficult to generalize the trained policies. However, by implementing the attention mechanism, our agents are capable of selectively focusing on relevant aspects of the environment. This focused attention helps maintain consistency in decision-making across different scenarios. Thus, the attention mechanism serves as a method for filtering information, which is crucial for achieving generalizability when interacting with previously unseen teammates.

## 3 Background

### 3.1 Markov Games

We consider multi-agent Markov Decision Processes (MDPs) (Littman, 1994), where the state transitions and rewards depend on the actions of all agents. Formally, a Markov game for $N$ agents is defined by a set of states $\mathcal{S}$, a set of actions $\mathcal{A}_i$ for each agent $i$, and a transition function $T : \mathcal{S} \times \mathcal{A}_1 \times \cdots \times \mathcal{A}_N \rightarrow \Delta(\mathcal{S})$, where $\Delta(\mathcal{S})$ denotes the set of probability distributions over $\mathcal{S}$. Each agent $i$ receives a reward as a function of the state and action of all agents, denoted by $R_i : \mathcal{S} \times \mathcal{A}_1 \times \cdots \times \mathcal{A}_N \rightarrow \mathbb{R}$. The goal of each agent is to maximize its expected discounted reward $E[\sum_{t=0}^{\infty} \gamma^t R_i(s_t, a_{1,t}, \ldots, a_{N,t})]$, where $\gamma \in [0, 1]$ is the discount factor and $s_t$ denotes the state at time $t$. In the multi-agent reinforcement learning (MARL) context, each agent typically learns a policy $\pi_i : \mathcal{S} \rightarrow \Delta(\mathcal{A}_i)$ that specifies the action distribution given the current state, aiming to optimize its own long-term payoff in the environment defined by the Markov game.

### 3.2 Gradient Field Representation in Multi-Agent Systems

The Gradient Field (GF) representation in multi-agent systems has proven to yield better policy learning (Long et al., 2024; Wu et al., 2022). Instead of using raw observation of the environment, the GF is a higher level representation which enhances the ability of agents in the environment. The key idea is that a Denoised Score Matching (DSM) (Song et al., 2020) generative model aims to learn the gradient field of a log-data-density; *i.e.*, the *score function*. Given samples $\{\mathbf{x}_i\}_{i=1}^N$ from an unknown data distribution $\{\mathbf{x}_i \sim p_{\text{data}}(\mathbf{x})\}$, the goal is to learn a *score function* to approximate $\nabla_{\mathbf{x}} \log p_{\text{data}}(\mathbf{x})$ via a *score network* $\mathbf{s}_\theta(\mathbf{x}) : \mathbb{R}^{|\mathcal{X}|} \rightarrow \mathbb{R}^{|\mathcal{X}|}$, by adopting an extension of DSM that estimates a *time-dependent score network* $\mathbf{s}_\theta(\mathbf{x}, t) : \mathbb{R}^{|\mathcal{X}|} \times \mathbb{R}^1 \rightarrow \mathbb{R}^{|\mathcal{X}|}$ to denoise the perturbed data from different noise levels simultaneously:

$$\mathcal{L}(\theta) = \mathbb{E}_{t \sim \mathcal{U}(\epsilon, T)} \left( \mathbb{E}_{\substack{\widetilde{\mathbf{x}} \sim q_{\sigma(t)}(\widetilde{\mathbf{x}}|\mathbf{x}) \\ \mathbf{x} \sim p_{\text{data}}(\mathbf{x})}} \lambda(t) \left\| \mathbf{s}_\theta(\widetilde{\mathbf{x}}, t) - \frac{1}{\sigma^2(t)}(\mathbf{x} - \widetilde{\mathbf{x}}) \right\|_2^2 \right), \tag{1}$$

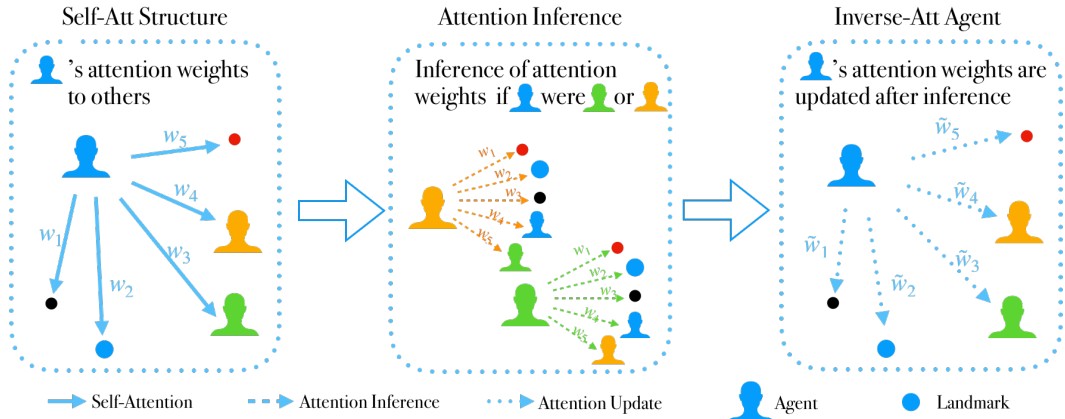

Figure 1: Pipeline for training the inverse attention agent: The first phase involves applying a self-attention mechanism, where the agent assigns attention weights to its observations and acts based on these weights. In the second phase, the agent performs attention inference on other agents of the same type using the inverse attention network. By placing itself in the position of these agents, it infers their attention weights, gaining insights into their goals and behaviors. In the final phase, the inverse attention agent uses the inferred information from the previous step to update its original attention weights, $\{w_1, w_2, \ldots, w_n\}$ to $\{\tilde{w}_1, \tilde{w}_2, \ldots, \tilde{w}_n\}$, consequently leading to changes in its final actions.

where $T$, $\epsilon$, $\lambda(t) = \sigma^2(t)$, $\sigma(t) = \sigma_0^t$, and $\sigma_0$ are hyper-parameters. The optimal time-dependent score network holds $\mathbf{s}_\theta^*(\mathbf{x}, t) = \nabla_{\mathbf{x}} \log q_{\sigma(t)}(\mathbf{x})$ where $q_{\sigma(t)}(\mathbf{x})$ is the perturbed data distribution:

$$q_{\sigma(t)}(\widetilde{\mathbf{x}}) = \int q_{\sigma(t)}(\widetilde{\mathbf{x}}|\mathbf{x}) p_{\text{data}}(\mathbf{x}) d\mathbf{x}. \tag{2}$$

When learning from different offline datasets $D_N$, $N$ gradient field (GF) representation functions $s_\theta$ are learned, resulting in the observation representations $s_{\theta 1}(o), s_{\theta 2}(o), \ldots, s_{\theta n}(o)$. These GF representations can be seen as the goals of the agent within specific environments, potentially offering greater effectiveness than raw observations such as relative coordinates. This is because GFs are not limited to specific objects and can more directly represent future trends, which are aligned with the agent's goals. By leveraging these learned representations, agents can gain a deeper understanding of their environment and make more informed decisions.

## 4   PROBLEM STATEMENT

In the context of a fully observable Multi-agent environment $E(N, \{A_i\})$ under the MDP settings, where agents types are fixed and can make observations (previous action included) of itself and also of other agents, we aim to learn a decentralized policy $\pi$. This policy, trained with a single group of agents, should achieve optimal long-term pay off not only with the trained agents but also when interacting with previously unseen agents.

## 5   METHOD

In this section, we introduce the inverse-attention agent, which acts based on its attention weights of goals and updates the weights based on the inferred goal weights of other agents. The agent is trained in three phases. The first phase involves applying the self-attention mechanism to the policy function (Vaswani et al., 2017) so that an agent acts based on the weights of the attentions. The second phase is to infer the attention of other agents of the same type using the inverse attention network. By placing itself in the position of the other agents, the agent infers their attention weights, gaining insights into their attention and behavior. The final phase involves training the inverse attention agent. The agent uses the inferred attention weights from the previous phase to update its own attention weights, consequently leading to changes in its final actions. The following subsections present the details of our algorithm. The overall pipeline is shown in Figure 1. The architecture of the inverse attention agent policy network is shown in Figure 2.

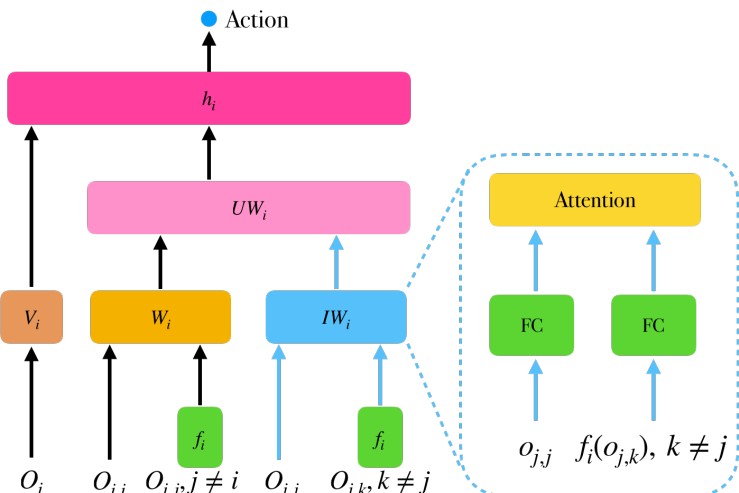

Figure 2: Network architecture of the inverse attention agent. For agent $i$, $W_i$ is the observation embedding function which takes in the observation and outputs initial attention weights. $IW_i$ is the inverse attention network which takes in the action and observation of the other agents and outputs the inferred attention weights. The $UW_i$ takes consideration of self initial weights and inferred weights from others and update $a_i$'s attention weights. The $h_i$ function outputs the final action based on the updated weights.

## 5.1 SELF-ATTENTION STRUCTURE

We incorporate a self-attention mechanism into the policy network in order to explicitly model the agent's mental states through attention weights assigned to different goals. These mental states are crucial, as they influence the agent's final action, and altering the attention weights will lead to changes in the action. Additionally, during this phase, we prepare the data required for training the inverse attention network. The agent trained with this structure is referred to as the Self-Att agent.

**Self-Attention Network:** Instead of using an MLP for the policy network, we adopt a structure similar to that proposed by Long et al. (2020), which utilizes attention embeddings of the goal for the decentralized policy network. This can be represented as

$$\pi_i(\mathbf{o_i}) = h_i(W_i(f_i(o_i)), V_i(f_i(o_i))), \quad o_i = o_{i,1}, o_{i,2}, \ldots, o_{i,N}, \tag{3}$$

where $o_i$ is the observation of agent $i$, which can be decomposed into a combination of $N$ goals within the environment, $f_i$ is a two-layer fully connected network that embeds the $N$ goals, $W_i(o_i)$ is the weight encoder that outputs the attention weights of the goals, whose details we will discuss later, and $V_i$ is the value function implemented as a two layer fully connected layer. The function $h_i$ multiplies the attention weights with all the goal information to get the weighted goals $goal_{\text{weighted}} = \sum w_{i,j} V_i(f_i(o_{i,j}))$ and then passed into a two-layer fully connected neural network to output the final action.

**Attention Weight Function:** We define the attention weight embedding function as

$$W_i(f_i(o_i)) = \text{attention}(f_i(o_{i,i}), f_i(o_{i,j})), \quad \forall j \neq i. \tag{4}$$

Self-attention is applied to the agent's own information along with all the goals and generates the output attention weights.

**Attention Inference Dataset Construction:** We collect the data pairs $(\{w_{i,1}, w_{i,2}, \ldots, w_{i,N}\}, o_i)$ during the training of the self-attention network. Once the policy has converged, we retain only the most recent $1/10$ of these data pairs in the dataset $D$ for training our inverse attention network.

## 5.2 ATTENTION INFERENCE

We train the inverse attention network using the attention inference dataset $D$. This network operates inversely to the self-attention network: It outputs the predicted attention weights based on the given goals and past actions. An agent then applies the inverse attention network to other agents of the same type to infer their attention weights. This allows the current agent to infer the attentions of other agents without knowing the ground truth values of their attention weights.

**Inverse Attention Network:** We propose the inverse attention network, designed to infer the attention of agents of the same type from their perspective. This function has the following structure:

$$\{w_{i,1}, w_{i,2}, dots, w_{i,N}\} = IW_i(o_i) = \text{attention}(o_{i,i}, f_i(o_{i,j})), \quad \forall j \neq i. \tag{5}$$

Similar to the attention weight function, the $IW_i$ function predicts the attention weight by applying self-attention between the embedding of self-information $o_{i,i}$ and the embedding of other goals $f_i(O_{i,j})$. Using the dataset $D$, which is agent's own data collected during the training process of Phase 1, we train the $IW_i$ network by minimizing the following loss function:

$$\text{Loss} = ||\{w_{i,1}, w_{i,2}, dots, w_{i,N}\}, \text{IW}_i(o_i)||. \tag{6}$$

**Inference of Others:** To estimate the attention weights of other agents, we position the current agent as if it were one of the other agents. In a fully observable environment, we gather the observations and previous actions of other agents of the same type. Then, we utilize the inverse attention network to infer their attention towards different goals. The inference attention weight for $a_j$ is

$$\{\tilde{w}_{j,1}, \tilde{w}_{j,2}, dots, \tilde{w}_{j,N}\} = IW_i(o_j). \tag{7}$$

## 5.3 INVERSE ATTENTION AGENT

We construct the inverse attention agent (Inverse-Att) by concatenating its original attention weights with the inferred weights. This concatenated vector is then passed through a fully connected layer, $UW_i$, to update the attention weights. This mechanism enables an inverse attention agent to adapt its attention weights and consequently adjust its actions based on the attentions of other agents. The process is represented by the following equation:

$$\{\tilde{w}_{i,1}, \tilde{w}_{i,2}, \ldots, \tilde{w}_{i,N}\} = UW_i(W_i(f_i(o_i)), IW_i(o_j)), \quad \forall j \neq i \text{ and } a_j \text{ is the same type as } a_i, \tag{8}$$

where $o_j$ is the observation from the other agent in the previous step, $IW_i(o_j)$ outputs the inverse attention estimation of agent $j$ from agent $i$, and $UW_i$ is the weight updating model, which concatenates the estimated attention weights from other agents of the same type with the original weights $W_i(f_i(x))$. The concatenated weights are then passed through a one-layer fully connected network to update the attention weights to $\tilde{w}$. Given that the attention weight embedding is a high-level representation, only a shallow network is required.

For fast convergence, the $UW_i$ are initialized to 1 when connected to the original attention weight and 0 when connected to the other agents' attention weights. This initialization guarantees that the initial output aligns with that of the Self-attention agent, maintaining unchanged attention weights. Commencing the optimization process from this point enhances effectiveness.

The complete policy network module for the inverse attention agent is outlined as follows:

$$\pi_i(\mathbf{o_i}) = h_i(UW_i(W_i(f_i(o_i)), IW_i(o_j)), V_i(f_i(x)))), \quad \forall j \neq i \text{ and } a_j \text{ is the same type as } a_i. \tag{9}$$

Algorithm 1 is the inverse attention algorithm.

## 6 EXPERIMENTS

We adopted MAPPO (Yu et al., 2022) as our MARL training scheme. We evaluated the adaptive performance of the policies in collaboration with human participants across all these tasks and found that the inverse attention agent performed well in these environments. Additionally, we compared our agents' behavior with human behavior. Our results indicate that the inverse attention agent is more adaptable to a wider variety of agents.

---

**Algorithm 1:** Inverse Attention Algorithm

---

**Data:** environment $E(N, \{A_i\})$ with $N$ agents
**Result:** an inverse attention agent
Use (3) as the policy function for the target training agent $a_i$;   `// self-att framework`
**while** *$a_i$'s policy network is not converged* **do**
    MARL training on $a_i$;
    Collect $\{w, o_i\}$ pair and store it to the dataset D;   `// training dataset for IW`
Keep the 1/10 latest $\{w, o_i\}$ in D and discard the rest;
Training inverse attention network *IW* using $\{w, o_i\}$ pair from the trimmed $D$ by following eq.6;
Replace the policy network of $a_i$ with (9) using trained *IW*; `// inverse-att framework`
**while** *$a_i$'s policy network is not converged* **do**
    MARL training on $a_i$;
**Return**: $a_i$ is the inverse attention agent;

---

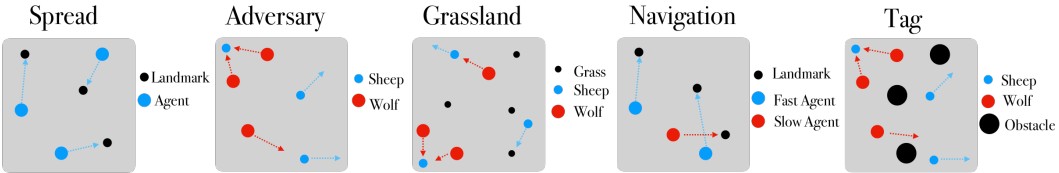

Figure 3: Environment visualization of the spread, adversary, and grassland games

## 6.1 ENVIRONMENTS

All of the environments used in our experiment were built on top of MPE (Mordatch and Abbeel, 2017; Lowe et al., 2017). Agents are depicted as particles inhabiting a continuous two-dimensional space, where they navigate, interact with one another, and with static landmarks, all within discrete time intervals. The key distinction lies in our application of the GF function atop raw observations, resulting in a *gf* representation structured as $\{gf_1, gf_2, \ldots, gf_N, gf_{\text{wall}}\}$. This representation comprises the gradient fields of $N$ entities and the gradient field of the boundaries, serving as the goals for the agents. Appendix A provides additional insights into the *gf* representation and the environment. We assessed the efficacy of our algorithm across five challenging environments: Spread, Adversary, Grassland, Navigation, and Tag on MPE. Visualizations of these environments are depicted in Figure 3.

**Spread:**   This is a fully cooperative game. In this scenario, $N$ agents must spread to cover $N$ landmarks. At each discrete time step, an agent earns 5 points if it occupies a landmark, and 0 points otherwise. The optimal state is each agent occupies a distinct landmark. The default setting is $N = 3$.

**Adversary:**   This is a fully competitive game that involves two types of agents: $N$ sheep and $N$ wolves. Wolves catch sheep, while sheep avoid capture. Sheep are faster than wolves. Each wolf earns 5 points for catching a sheep, while each caught sheep loses 5 points. The scale of this game is denoted as $N - N$. The default setting is $N = 3$.

**Grassland:**   This is a mixed game. Entities include $N$ sheep, $N$ wolves, and 4 grass. Sheep aim to collect as much grass as possible while evading wolves. Wolves seek to catch as many sheep as they can. Each sheep earns 3 points for each collected grass but loses 5 points for each capture. Wolves earn 5 points for each caught sheep. Grass re-spawns at random positions after being consumed by sheep. The scale of this game is denoted as $N - N$. The default setting is $N = 3$.

**Navigation:**   This is a fully cooperative game with a group reward system. There are two fast agents and one slow agent, and the goal is for the agents to navigate to three different landmarks as quickly as possible. The team earns 5 points for each landmark occupied, shared equally among all agents. The fast agents must accommodate the slow agent to help reach the more distant landmarks.

Table 1: Full results in all the tested environments

|  | MAPPO | IPPO | MAA2C | ToM2C* | Self-Att | Inverse-Att |
|---|---|---|---|---|---|---|
| Spread | 31.82±1.21 | 0.55±0.01 | 48.46±0.47 | 49.04±3.77 | 283.89±6.23 | **404.14±5.40** |
| Adversary Sheep | -113.97±1.74 | -121.03±4.17 | -96.54±3.26 | -86.15±2.80 | -23.15±0.83 | **-16.91±1.84** |
| Adversary Wolf | 48.76±1.86 | 25.18±1.40 | 109.14±0.69 | 61.58±1.31 | 107.93±2.26 | **110.15±3.74** |
| Grassland Sheep | -84.50±4.20 | -86.54±0.32 | -70.79±2.41 | -61.60±1.98 | -10.02±5.19 | **28.59±0.93** |
| Grassland Wolf | 27.86±0.79 | 22.60±1.38 | 74.28±1.27 | 41.11±1.59 | 93.68±1.61 | **101.21±2.84** |
| Navigation | 251.14±4.93 | 230.10±0.89 | 279.87±7.05 | 249.41±5.29 | 328.24±7.76 | **497.96±10.75** |
| Tag Sheep | -93.51±2.13 | -111.79±3.03 | -77.81±2.03 | -56.69±1.02 | -11.74±0.65 | **-5.99±0.44** |
| Tag Wolf | 42.17±3.46 | 26.77±0.82 | 59.41±1.10 | 52.99±1.17 | 67.60±1.00 | **109.81±1.36** |

**Tag:** This is a mixed game involving 3 wolves and 3 sheep, with group rewards. The objective for the wolf team is to catch the sheep while avoiding three obstacles during the chase. When any wolf catches a sheep, all wolves received 5 points and all sheep lose 5 points.

## 6.2 BASELINE AND EVALUATION METHODS

In the experiments, we compared agents trained with the following methods:

1. **MAPPO:** We follow the same method as (Yu et al., 2022).
2. **IPPO:** We follow the same method as (Schulman et al., 2017)
3. **MAA2C:** It extends the existing on-policy actor-critic algorithm A2C (Mnih et al., 2016) by applying centralised critics conditioned on the state of the environment.
4. **ToM2C*:** A modified ToM method adapted from (Wang et al., 2022), where agents directly predict other agents goals without communication.
5. **Self-Att:** We adopt the self-attention structure proposed in Section 5.1.
6. **Inverse-Att:** Our proposed attention inference agent described in Section 5.3.

All the baseline methods are trained for the same amount of accumulative episodes as an Inverse-Att agent required. All agents trained in these methods do not see agents of other methods until evaluation time.

We assessed the performance of our algorithm through *mix-and-match*. For each method, we select three different seeds to form three distinct groups. We formed an agent pool by putting agents trained using all the evaluated methods. During the cross-competition period, agents were randomly sampled from this pool to form a unique composition. The cross competitions were conducted over 1000 episodes with each episode consisting of 200 steps. During each episode, we recorded the agents' rewards according to their respective methods and calculated the average reward for performance comparison.

## 6.3 QUANTITATIVE RESULTS

In this section, we delve into the quantitative results across all five games. We conducted training for MAPPO, IPPO, MAA2C, ToM2C* and Self-Att agents over 40 million steps across all scales. For training the Inverse-Att agent, Phase 1 encompassed 20 million steps, followed by another 20 million steps for Phase 3. Phase 2 training involved offline learning, obviating the need for interaction with the environments. Subsequently, evaluations were carried out over $2 \times 10^6$ steps of mix-and-match for all five games.

We tested all the methods across all environments, with the full results presented in Table 1. The Inverse-Att method demonstrates significant improvement over all the other methods across the five games in cooperative, competitive, and mixed settings, as well as with both individual and team rewards. This highlights the superior adaptability and generality of Inverse-Att when dealing with unseen agents. Self-Att ranks second, while the remaining baselines perform similarly and rank below both Inverse-Att and Self-Att. Due to the similar performance of MAPPO, IPPO, MAA2C, and ToM2C*, and computational limitations, in the following sections we will focus on MAPPO, Self-Att, and Inverse-Att in the Spread, Adversary, and Grassland games.

Table 2: Spread results

|   | MAPPO | Self-Att | Inverse-Att |
|---|---|---|---|
| 2 | $51.14 \pm 1.15$ | $288.75 \pm 2.80$ | $\mathbf{363.02 \pm 1.99}$ |
| 3 | $31.10 \pm 0.93$ | $258.81 \pm 5.35$ | $\mathbf{379.95 \pm 1.99}$ |
| 4 | $19.59 \pm 0.22$ | $253.48 \pm 1.13$ | $\mathbf{269.12 \pm 0.86}$ |

Table 3: Adversary results

|   | MAPPO WOLF | Self-Att WOLF | Inverse-Att WOLF | MAPPO SHEEP | Self-Att SHEEP | Inverse-Att SHEEP |
|---|---|---|---|---|---|---|
| 2-2 | $14.30 \pm 0.61$ | $39.75 \pm 1.88$ | $\mathbf{42.45 \pm 1.50}$ | $-64.23 \pm 1.70$ | $-18.49 \pm 0.48$ | $\mathbf{-13.82 \pm 1.10}$ |
| 3-3 | $29.30 \pm 1.42$ | $71.95 \pm 2.38$ | $\mathbf{75.44 \pm 1.02}$ | $-132.75 \pm 2.27$ | $-30.79 \pm 1.82$ | $\mathbf{-12.61 \pm 0.24}$ |
| 4-4 | $32.93 \pm 1.21$ | $66.75 \pm 1.62$ | $\mathbf{67.15 \pm 0.97}$ | $-134.31 \pm 3.01$ | $-16.82 \pm 0.28$ | $\mathbf{-15.59 \pm 0.65}$ |

Table 4: Grassland results

|   | MAPPO WOLF | Self-Att WOLF | Inverse-Att WOLF | MAPPO SHEEP | Self-Att SHEEP | Inverse-Att SHEEP |
|---|---|---|---|---|---|---|
| 2-2 | $20.80 \pm 0.23$ | $36.85 \pm 2.18$ | $\mathbf{56.69 \pm 1.78}$ | $-70.17 \pm 1.71$ | $10.25 \pm 2.33$ | $\mathbf{33.66 \pm 1.23}$ |
| 3-3 | $22.06 \pm 0.78$ | $65.50 \pm 1.96$ | $\mathbf{70.28 \pm 0.78}$ | $-99.73 \pm 1.19$ | $-21.52 \pm 1.32$ | $\mathbf{25.45 \pm 0.13}$ |
| 4-4 | $35.38 \pm 0.54$ | $46.35 \pm 0.79$ | $\mathbf{81.67 \pm 1.51}$ | $-138.71 \pm 2.68$ | $14.14 \pm 0.71$ | $\mathbf{36.16 \pm 0.14}$ |

## 6.4 IMPACT OF DIFFERENT POPULATION SCALES

We tested the scalability of Inverse-Att by comparing with MAPPO and Self-Att in the Spread, Adversary, and Grassland games in the MPE environment.

In the Spread game, evaluations were conducted across three scales: 2, 3, and 4 agents. The results are detailed in Table 2. Meanwhile, in the Adversary and Grassland games, evaluations were conducted across scales of 2-2, 3-3, and 4-4 for both sheep and wolves. The outcomes are presented in Table 3 and Table 4, respectively.

Across all three games, notable enhancements were evident when comparing the outcomes of MAPPO and Self-Att, underscoring the efficacy of the self-attention network. The Inverse-Att method exhibited superior performance across the tested environments, particularly in cooperative-related games such as Spread and Grassland. This superiority likely stems from the attention inference being exclusively applied to agents of the same type (teammates), a factor more critical in cooperative settings than in fully competitive games.

## 6.5 HUMAN EXPERIMENTS

To further assess the adaptive capabilities of the Inverse-Att agents, we conducted human experiments to evaluate their performance in collaboration with human players. Five participants engaged in Spread, Grassland and Adversary games across the following scales: Spread (3 agents), Adversary (3-3), and Grassland (3-3), assuming five distinct roles (agent for Spread; sheep and wolf for Adversary and Grassland). In each role, participants cooperated with teammate agents of the same type trained using each of the three methods (MAPPO, Self-Att, Inverse-Att) over five episodes, while opponent agents remained consistently MAPPO agents. Agents were sampled by type in each environment and then fixed to ensure uniformity across participant groups. Each episode comprised 100 steps, with rewards recorded based on the respective training methods. The results are summarized in Table 5. Appendix A provides further insights.

It is noteworthy that both Self-Att and Inverse-Att agents outperformed human participants in the tested environments. Across most roles within the environments, Inverse-Att agents exhibited greater adaptability to human players and demonstrated greater stability compared to the other methods, with exceptions observed in the role of wolf in the Grassland environment.

Table 5: Results of the human experiments

|  | MAPPO | Self-Att | Inverse-Att | Human |
|---|---|---|---|---|
| Spread | $16.8 \pm 13.81$ | $272.0 \pm 30.41$ | **$332.3 \pm 17.13$** | $115.86 \pm 55.89$ |
| Adversary (Wolf) | $81.5 \pm 38.42$ | $197.4 \pm 25.85$ | **$286.9 \pm 22.21$** | $188.4 \pm 74.87$ |
| Adversary (Sheep) | $-48.8 \pm 24.06$ | $-8.0 \pm 3.46$ | **$-0.8 \pm 1.16$** | $-15.33 \pm 10.47$ |
| Grassland (Wolf) | $49.3 \pm 20.38$ | **$197.9 \pm 12.76$** | $185.7 \pm 30.45$ | $132.86 \pm 51.84$ |
| Grassland (Sheep) | $-20.75 \pm 13.95$ | $9.8 \pm 8.66$ | **$20.34 \pm 2.33$** | $-30.52 \pm 32.15$ |

Table 6: Group reward in spread game with multi Inverse-Att agents

| | #$N$ Inv-Att | | | | |
|---|---|---|---|---|---|
| Scale | 0 | 1 | 2 | 3 | 4 |
| 2 | $86.44 \pm 0.36$ | $472.92 \pm 4.93$ | $593.57 \pm 7.02$ | - | - |
| 3 | $148.53 \pm 1.64$ | $635.40 \pm 7.01$ | $812.77 \pm 5.12$ | $1070.28 \pm 8.75$ | - |
| 4 | $109.26 \pm 1.60$ | $494.59 \pm 11.30$ | $701.60 \pm 13.95$ | $818.24 \pm 8.20$ | $1140.23 \pm 4.67$ |

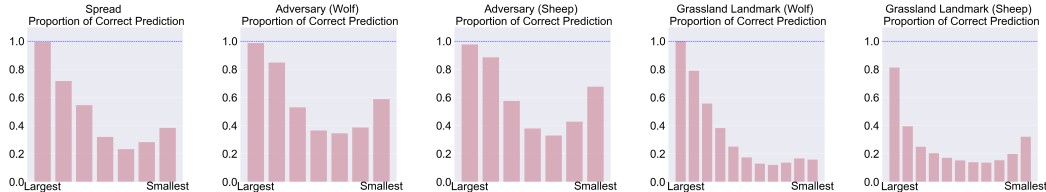

Figure 4: We evaluated the prediction accuracy of the inverse attention network across five roles in the spread, adversary, and grassland environments under the scale of {spread: $3$, adversarial: $3 - 3$ and grassland: $3 - 3$}. In each bar graph, from left to right, we display the prediction accuracy from the most attended goal to the least attended goal. The results demonstrate that the inverse network can accurately predict the attentions of other agents, particularly for the top two attentions of interest.

### 6.6 IMPACT OF MULTIPLE INVERSE-ATT AGENTS

In multi-agent systems, incorporating Theory of Mind (ToM) can introduce intricate dynamic interactions. This complexity is amplified when the inference is reciprocal, potentially creating a cognitive loop that exacerbates uncertainty and leads to unpredictable or suboptimal decision-making outcomes. Our objective is to explore the impact of increasing the number of Inverse-Att agents on emergent behavior patterns.

We initiated the investigation by replacing randomly sampled MAPPO agents, trained with different seeds, with Inverse-Att agents gradually. Evaluations were conducted over $2 \times 10^6$ steps per group in the Spread game, across three different scales: 2, 3, and 4 agents. The total rewards earned by the teams are summarized in Table 6. We observe a nonlinear marginal return pattern as more Inverse-Att agents are introduced into the game at all three scales. This observation underscores the effectiveness of our Inverse-Att agent in cooperating effectively with other attention-aware agents.

### 6.7 INVERSE ATTENTION NETWORK PREDICTION ACCURACY

We collected $2 \times 10^6$ steps of weights-observation pair from one Self-Att agent per environment, considered as the attention ground truth of those agents. We then inputed the observation into our inverse attention network and compared the predicted weights with the ground truth across the spread, adversarial, and grassland environments at scales {spread: $3$, adversarial: $3 - 3$ and grassland: $3 - 3$}. We evaluated the effectiveness of our inverse network in accurately predicting weights based on their rank. For example, if the inverse network identifies $gf_{\text{wall}}$ as having the highest weight, this prediction is considered accurate if the self-attention network also determines that the weight of $gf_{\text{wall}}$ is the highest. The results are summarized in Figure 4. The prediction of the most significant attention

reaches nearly 100% accuracy, demonstrating that the inverse network can accurately predict the attentions of other agents, especially for the top two attentions of interest.

## 7 CONCLUSIONS

We have introduced the Inverse Attention Agent, which operates by leveraging attention weights to infer the attention of other agents. It then utilizes these inferred weights to adjust its own attention weights, thereby fine-tuning its final actions. The Inverse Attention Agent demonstrates superior adaptability to unseen agents, as evidenced by interactions with policies trained using various methods, as well as with human participants.

### 7.1 LIMITATIONS AND FUTURE WORK

Currently, the attention inference is limited to the same type of agents. In future work, we would like to model the Theory of Mind of agents of different types. We will also develop an attention model for the *UW* network that can accommodate an arbitrary number of inferred attention weights.

## A APPENDIX

### A.1 ENVIRONMENTAL DETAILS

**Reward:** In the Spread scenario, during training, agents are awarded a $+100$ reward for occupying a landmark at every timestep. For reward engineering, the agent's reward will be deducted proportionally to the minimum distance to the landmarks $R_{\text{distance}} = 0.2 \min(\text{Distance}(\text{self}, \text{Landmark}_{\text{all}}))$. In the Adversary scenario, wolves are awarded with a $+100$ reward for every sheep they catch, while sheep are awarded with a $-100$ reward for every time they are caught. The wolf's reward will be deducted proportionally to the minimum distance to the sheep $R_{\text{distance}} = 0.2 * \min(\text{Distance}(\text{self}, \text{Sheep}_{\text{all}}))$. In the Grassland scenario, wolves are awarded a $+5$ reward for every sheep they catch, while sheep are awarded a $-5$ reward for every time they are caught and a $+2$ reward for every landmark they occupy. The wolves' reward will be deducted proportionally to the minimum distance to the sheep $R_{\text{distance}} = 0.2 \min(\text{Distance}(\text{self}, \text{Sheep}_{\text{all}}))$, whereas the sheep's reward will be deducted proportionally to the minimum distance to the landmarks $R_{\text{distance}} = 0.2 \min(\text{Distance}(\text{self}, \text{Landmark}_{\text{all}}))$. For the Navigation environment, the group reward is defined as a $+5$ reward for every landmark an agent occupies. For the Tag environment, when any wolf catches a sheep, all wolves receive a $+5$ and all sheep receive a $-5$ reward.

**Observation:** In the Spread scenario, *MAPPO* and *Self-Att* agents receive information regarding their own position and velocity, the positions of other agents, and the positions of landmarks. They use these information to generate the gradient field $\{\text{velocity}, gf_o\}$ as the observation. *Inverse-Att* agents also receive the past actions and past observations of their teammates and have { teammate past action, teammate past observation } in addition to the gradient fields. In the Adversary scenario, the observation is the same except for the omission of $gf_{\text{landmarks}}$, and Grassland, Navigation, and Tag have exactly the same observation space.

**Action:** The agent's action is represented by a two-dimensional continuous vector, which describes the force applied to an entity, considering both magnitude and angular direction.

### A.2 TRAINING DETAILS

Table 7 presents the hyperparameters for the gradient field. Table 8 and Table 9 present the hyperparameters for training all the agents.

70% of the dataset is used for training, 10% of the dataset is split into a validation set, which is used for early stopping, and 20% of the dataset is used for testing.

Table 7: Hyperparameters for the gradient field

|  | lr | sigma | $t_0$ | hidden size | optimizer | optimizer betas | network |
|---|---|---|---|---|---|---|---|
| Agent GF | 2e-4 | 25 | 1 | 64 | Adam | [0.5, 0.999] | GNN |
| Boundary GF | 2e-4 | 25 | 1 | 64 | Adam | [0.5, 0.999] | MLP |

Table 8: Hyperparameters for the agent training

|  | lr | gain | share policy | optimizer | critic lr | ppo epoch |
|---|---|---|---|---|---|---|
| MAPPO | 7e-4 | 0.01 | False | Adam | 7e-4 | 10 |
| IPPO | 7e-4 | 0.01 | False | Adam | 7e-4 | 10 |
| MAA2C | 7e-4 | 0.01 | False | Adam | 7e-4 | 10 |
| ToM2C* | 7e-4 | 0.01 | False | Adam | 7e-4 | 10 |
| Self-Att | 7e-4 | 0.01 | False | Adam | 7e-4 | 10 |
| Inverse-Att | 7e-4 | 0.01 | False | Adam | 7e-4 | 10 |

Table 9: Hyperparameters for the inverse network

| lr | batch size | hidden dim | patience threshold | num epoch | Optimizer |
|---|---|---|---|---|---|
| 0.001 | 64 | 64 | 100 | 3000 | Adam |

### A.3 GRADIENT FIELD SYNTHETIC DATA GENERATION

We use synthetically-generated data to train the Agent Gradient Field as well as the Boundary Gradient Field. Synthetic data generation is performed as follows:

**Entity Gradient Field:** We randomly generated 10,000 two-dimensional points within a $2 \times 2$ grid as one entity's position. We then randomly generated another entity's position close to the previous position such that the L1 distance is less than $10^{-5}$. We mark this *gf* representation as $gf_e$. This *gf* function takes in the agent's location and the relative position of another entity.

**Boundary Gradient Field:** We randomly generated 10,000 positions $(x, y) \in [-0.8, 0.8] \times [-0.8, 0.8]$. We mark this *gf* representation as $gf_{\text{wall}}$, which takes in only the agent's position.

### A.4 GRADIENT FIELD REPRESENTATION OF THE ENVIRONMENT

For the spread, adversary, and grassland environments, we applied the entity gradient field for all the other entities around that agent as entity attentions, as well as a boundary gradient field for the environment attention. Thus, we transformed the observation from $\{o_{i,1}, o_{i,2}, \ldots, o_{i,n}\}$, which is the information of $N$ entities, to $\{gf_e(o_{i,1}), gf_e(o_{i,2}), \ldots, gf_e(o_{i,n}), gf_{\text{wall}}(o_{i,i})\}$.

### A.5 QUALITATIVE RESULTS

Figure 5 provides example screenshots of a representative match involving *Inverse-Att* agents in the Spread scenario. In these images, the blue balls represent *Inverse-Att* agents, while the small black balls are landmarks. The left figure shows the initial state of all the agents. The middle figure demonstrates the *Inverse-Att* agents successfully navigating to their respective landmarks. The right figure shows the final result, where all three *Inverse-Att* agents occupy their own landmarks without conflict.

Additionally, we visualize two representative games in the Adversary and Grassland scenarios. In these visualizations, the smallest black balls are landmarks. The large balls represent wolves, with red indicating *Self-Att* agents and white indicating *Inverse-Att* agents. The small balls represent sheep, with blue indicating *Self-Att* sheep and white indicating *Inverse-Att* sheep. The screenshots illustrate

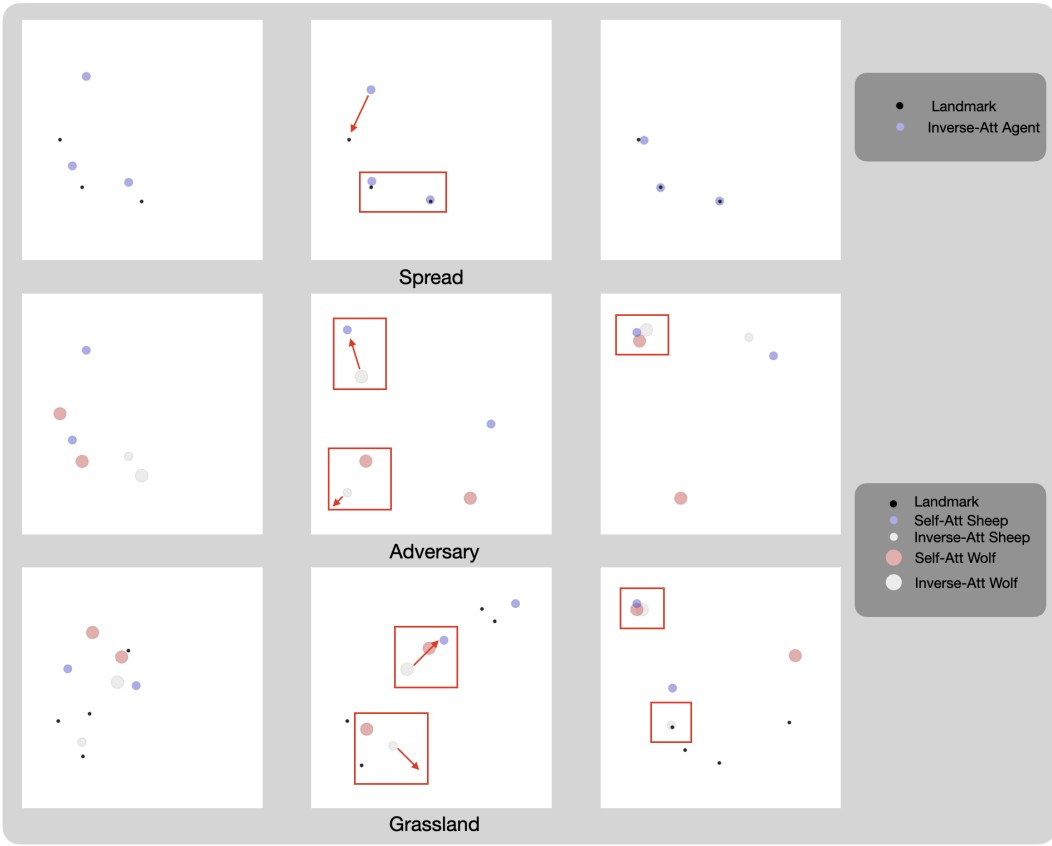

Figure 5: Qualitative results in the spread, adversary, and grassland games in MPE demonstrate that the Inverse-Att agents can successfully adapt to unseen agents.

Table 10: Results of the grassland game in the 10-10 scale environment

|  | MAPPO | IPPO | MAA2C | ToM2C* | Self-Att | Inverse-Att |
|---|---|---|---|---|---|---|
| Grassland Sheep | -68.15 | -74.64 | -55.00 | -58.96 | -143.50 | **-52.34** |
| Grassland Wolf | 90.47 | 49.81 | 30.5 | 30.21 | 120.08 | **132.39** |

that *Inverse-Att* wolves cooperate with unknown *Self-Att* agents to corner the sheep. *Inverse-Att* sheep successfully avoid wolves and capture as many landmarks as possible.

These results suggest that *Inverse-Att* agents can adapt successfully to new teammates, indicating that our approach is an effective method for creating adaptive agents that excel when paired with various types of teammates and opponents.

## A.6    SCALABILITY ANALYSIS

In Table 10 we present the results for 20 agents in the Grassland environment with a 10-10 scale. The Inverse-Att agents scale effectively and also outperform other ablations and baseline methods.

## A.7    PARTIAL OBSERVATION ANALYSIS

This approach can be extended to partial observations by applying inverse attention only to agents within the observable range. Since the inverse attention network leverages the attention mechanism, it is more robust to a dynamic number of inputs. We demonstrate the impact of partial observation by testing different visibility radii, 1.5, 1.0, and 0.5, in all the tested games (spread, adversary, grassland,

Table 11: Results of the grassland game in the 10-10 scale environment

| Radius=1.5 | MAPPO | IPPO | MAA2C | ToM2C* | Self-Att | Inverse-Att |
|---|---|---|---|---|---|---|
| Spread | 35.95 | 0.50 | 48.90 | 36.96 | 178.03 | **263.53** |
| Adversary Sheep | -62.63 | -72.94 | -66.36 | -59.14 | -19.72 | **-17.25** |
| Adversary Wolf | 40.83 | 13.83 | 53.56 | 40.92 | 57.64 | **94.18** |
| Grassland Sheep | -54.99 | -73.85 | -58.71 | -59.48 | -12.24 | **15.91** |
| Grassland Wolf | 35.94 | 18.87 | 64.82 | 27.78 | 69.40 | **97.21** |
| Navigation | 317.39 | 222.81 | 307.55 | 300.87 | 409.54 | **467.17** |
| Tag Sheep | -58.12 | -100.08 | -56.37 | -50.72 | -13.18 | **-6.93** |
| Tag Wolf | 38.66 | 18.25 | 49.21 | 45.46 | 66.83 | **70.5** |
| Radius=1 | MAPPO | IPPO | MAA2C | ToM2C* | Self-Att | Inverse-Att |
| Spread | 44.90 | 0.66 | 53.72 | 40.47 | 78.22 | **111.40** |
| Adversary Sheep | -37.92 | -58.72 | -60.51 | -46.10 | -21.97 | **-20.27** |
| Adversary Wolf | 42.79 | 14.01 | 39.68 | 32.66 | 39.62 | **77.37** |
| Grassland Sheep | -45.45 | -66.14 | -57.12 | -54.01 | **-2.26** | -20.41 |
| Grassland Wolf | 38.60 | 19.70 | 67.20 | 29.24 | 69.23 | **74.85** |
| Navigation | 267.76 | 162.79 | 262.10 | 237.79 | 243.65 | **269.83** |
| Tag Sheep | -42.62 | -105.53 | -50.67 | -51.19 | -16.01 | **-8.13** |
| Tag Wolf | 38.63 | 16.59 | 37.14 | 40.14 | 64.57 | **79.42** |
| Radius=0.5 | MAPPO | IPPO | MAA2C | ToM2C* | Self-Att | Inverse-Att |
| Spread | 53.28 | 0.60 | 22.73 | 47.36 | 19.22 | **59.16** |
| Adversary Sheep | -16.28 | -39.25 | -52.27 | -26.90 | -19.50 | **-13.47** |
| Adversary Wolf | 34.96 | 11.86 | 25.16 | 19.73 | 23.90 | **51.27** |
| Grassland Sheep | -34.30 | -55.67 | -53.54 | -39.50 | -23.32 | **-11.73** |
| Grassland Wolf | 40.87 | 21.57 | 50.94 | 28.99 | 57.40 | **61.80** |
| Navigation | 163.95 | 146.04 | 162.58 | 226.77 | 253.06 | **256.57** |
| Tag Sheep | -20.69 | -93.58 | -36.80 | -53.28 | -14.03 | **-6.86** |
| Tag Wolf | 30.28 | 15.57 | 54.98 | 29.60 | 39.51 | **58.07** |

Table 12: Result of Ablation of the GFs

| | MAPPO | IPPO | MAA2C | ToM2C* | Self-Att | Inverse-Att-Vanilla |
|---|---|---|---|---|---|---|
| Spread | 27.24 | 0.65 | 43.21 | 27.38 | 254.04 | **348.66** |
| Adversary Sheep | -101.54 | -118.97 | -106.39 | -104.60 | -24.56 | **-19.42** |
| Adversary Wolf | 34.91 | 16.99 | 90.17 | 51.44 | 90.86 | **95.17** |
| Grassland Sheep | -76.60 | -92.61 | -73.03 | -81.51 | -1.01 | **21.50** |
| Grassland Wolf | 24.23 | 14.85 | 79.99 | 24.46 | 100.66 | **103.05** |

Table 13: Comparison between Inverse-Att-Vanilla and Inverse-Att agents

| | Inverse-Att-Vanilla | Inverse-Att |
|---|---|---|
| Spread | 348.66 | **404.14** |
| Adversary Sheep | -19.42 | **-16.91** |
| Adversary Wolf | 95.17 | **110.15** |
| Grassland Sheep | 21.50 | **28.59** |
| Grassland Wolf | **103.05** | 101.21 |

navigation, tag). Across all three settings, agents using inverse attention consistently achieve the best performance. The results are shown in Table 11.

## A.8    ABLATION STUDY OF THE GRADIENT FIELD

Inverse-Att features two novel modules: the gradient field and the inverse attention network. Self-Att is an ablation of the inverse attention network, which is included as a baseline in the main paper. As shown in Table 1, Table 2, Table 3, Table 4, and Table 5, it consistently outperforms all the other baseline models.

The ablation study for Inverse-Att without the gradient field is extensively discussed in (Long et al., 2024). Additionally, we conducted experiments in the Spread 3-agents game, Adversary 3-3 game, and Grassland 3-3 game, replacing the gradient field with relative positions. This variant is referred to as Inverse-Att-Vanilla.

In our evaluation, Inverse-Att-Vanilla competed against the same set of baseline agents as Inverse-Att: MAPPO, IPPO, MAA2C, ToM2C, and Self-Att. For Inverse-Att-Vanilla agents, the input consisted solely of raw environmental observations, while all other agents retained gradient field inputs.

The results of these experiments are presented in Table 12, which details the performance of models competing with Inverse-Att-Vanilla. Furthermore, Table 13 compares the performance of Inverse-Att-Vanilla and Inverse-Att against the same set of agents in identical agent pools. The findings demonstrate that while Inverse-Att delivers strong performance compared to other baselines, it outperforms all other agents in most cases. This highlights the critical importance of incorporating gradient field representations into the observation space.

## A.9    HUMAN EXPERIMENT DETAILS

There are five experiments in total: Spread, Adversary (human plays wolf), Adversary (human plays sheep), Grassland (human plays wolf), and Grassland (human plays sheep). In each experiment, human players will cooperate with three types of agents, *MAPPO*, *Self-Att*, and *Inverse-Att*, for five rounds. The episode rewards are averaged for these rounds. Human players are not informed about the type of agents with which they are playing. In the Adversary and Cooperate games, the opponents are always *MAPPO* agents.

Five participants (4 males and 1 female) participated in this experiment. All were between the ages of 21 and 28 with normal or corrected-to-normal visual acuity. All participants were given the Experiment Instructions below and received rewards (free food) for their participation. All participants were instructed to use the UP, DOWN, LEFT, and RIGHT keys on the keyboard to control movements.

---

**Experiment Instructions to the Participants**

Thank you for participating in this research experiment designed to evaluate the performance of various MARL agents in adapting to human players. There are three scenarios: Spread, Adversary, and Grassland. In the Adversary and Grassland, you will play both as wolf and sheep. In total there are five scenarios to play. In each scenario, you will cooperate with three types of MARL agent, and you will play five rounds with each type of agents.

**Detailed Game Descriptions:**
- **Spread Game**:
    - *Setup*: The game area contains several small black balls, each representing a landmark. You are represented by a white ball, and your teammates by blue balls.
    - *Objective*: Each player must navigate to occupy a unique landmark. The number of landmarks equals the number of players.
    - *Gameplay*: Using directional controls, navigate towards the landmarks. Coordination with teammates might be necessary to ensure all landmarks are covered.
    - *Scoring*: You earn 5 points for every timestep you occupy a landmark.

---

- **Adversary Game**:
  - *Setup*: In this scenario, you are assigned the role of either sheep or wolves. The color of your ball is always white, with large ball indicating wolves and small ball indicating sheep. Large red balls are MARL wolves and small blue balls are MARL sheep.
  - *Objective*:
    * As a Sheep: Avoid the wolves.
    * As a Wolf: Work together with other wolves to catch the sheep.
  - *Gameplay*: Sheep must use speed and agility (as they move faster) to escape from wolves, while wolves need to coordinate their movements to catch sheep.
  - *Scoring*:
    * As a Sheep: Each wolf catch results in a deduction of 5 points from your score.
    * As a Wolf: You gain 5 points for every sheep you catch.

- **Grassland Game**:
  - *Setup*: The game environment includes the same players and roles as the Sheep Wolf Game, but now includes four small black balls (landmarks) that appear randomly in the game area.
  - *Objective*:
    * As a Sheep: Collect as many landmarks as possible while avoiding wolves.
    * As a Wolf: Catch the sheep while they attempt to collect landmarks.
  - *Gameplay*: Sheep must balance between quickly moving towards landmarks and evading wolves. Wolves, while trying to catch sheep, must also strategize to guard landmarks indirectly, making them risky spots for sheep.
  - *Scoring*:
    * As a Sheep: Gain 3 points for each landmark collected and lose 5 points each time you are caught by a wolf.
    * As a Wolf: Gain 5 points for each sheep caught. Landmarks disappear once collected and respawn randomly.

**Instructions for Each Turn:**

- At the start of each round, your role (sheep or wolf) and the specific game scenario will be communicated.

- Use the directional controls to navigate your character according to the game's objectives. You can move in four directions: up, down, left, and right.

- The games are within a $2 \times 2$ grid. You will not be able to move out of this grid.

- The game ends until the time expires.

## A.10 COMPUTATIONAL RESOURCES

**Hardware specifications:** All experiments are run on servers/workstations with the following configurations:

- 128 CPU cores, 692GB RAM
- 128 CPU cores, 1.0TB RAM
- 32 CPU cores, 120GB RAM
- 24 CPU cores, 80GB RAM, 1 NVIDIA 3090 GPU

All experiments can be run on a single server with 24 CPU cores, 80GB RAM, 1 NVIDIA 3090 GPU.

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
