# OpenReview forum: "Inverse Attention Agents for Multi-Agent Systems"
_ICLR.cc/2025/Conference — ICLR 2025 Poster_

### Official Review · Reviewer_HVvH · 2024-10-17

**Soundness:** 3
**Presentation:** 3
**Contribution:** 3
**Rating:** 6
**Confidence:** 3

**Summary:**

This paper introduces Inverse Attention Agents for multi-agent systems (MAS), addressing the challenge of dynamically adapting to diverse environments with unfamiliar teammates and opponents. Traditional agents struggle to perform well outside of their training environments, while this method leverages Theory of Mind (ToM) combined with attention mechanisms to enhance agent interactions. The core contribution is the development of an Inverse Attention Network, which allows agents to infer the attention weights and intentions of others, enabling more flexible cooperation and competition. The method is validated through extensive experiments in cooperative and competitive tasks using the Multi-agent Particle Environment (MPE), where the proposed agents demonstrate superior performance, particularly in ad-hoc teaming and human-agent cooperation scenarios.

**Strengths:**

1) The integration of Theory of Mind with attention mechanisms in MAS is novel, offering a new method for enhancing agent adaptability to unfamiliar teammates and opponents.
2) The method is well-justified theoretically, and the experimental setup is thorough, covering a range of environments and both competitive and cooperative tasks.
3) The paper is clearly written, and the presentation of the methodology, particularly the detailed description of the inverse attention network, is well-done.
4) The ability to improve agent cooperation in ad-hoc settings is an important problem in MAS, and this work provides a promising solution. The method also shows potential for applications beyond the tested environments, such as human-agent interaction in robotics.

**Weaknesses:**

1. The method is currently only tested in relatively simple MPE environments. While the results are promising, the approach’s applicability to more complex real-world scenarios (e.g., autonomous driving, large-scale multi-agent simulations) is not well explored.
2. As the number of agents increases, the computational complexity of inferring attention weights for multiple agents may become a bottleneck. The paper does not fully address how the method scales with a larger number of agents or more diverse agent types.
3. The method assumes that agents are of the same type, which limits its generalization. In many real-world applications, interactions between heterogeneous agents (with different abilities or goals) are common, but the paper does not address this scenario.
4. The inverse attention mechanism is trained in an offline manner, which may limit its adaptability in rapidly changing environments where real-time adjustments are necessary.

**Questions:**

1. Does the performance of inverse attention agents degrade as the number of agents increases, particularly in larger environments like the 4-4 Grassland game?

2. Why does the cooperative gain between multiple inverse attention agents not increase linearly, and is this due to cognitive loops or mutual inference inaccuracies?

3. Why do inverse attention agents perform inconsistently when cooperating with humans in some roles (e.g., sheep in Adversary), and how can this cooperation be made more robust?

4. How is the accuracy of the inverse attention network related to task success, and why doesn't high inference accuracy always correlate with better performance in tasks like Adversary?

5. What are the main factors behind the performance differences between inverse attention agents and baseline algorithms, and how can these differences be better explained?

6. Does the inverse attention mechanism perform better in competitive tasks than in cooperative ones, and how can its adaptability in complex cooperation tasks be improved?

---

> ### Author Response · Authors · 2024-11-23
> **Response to Reviewer HVvH**
>
> We thank you for your insightful comments on improving our paper. We hope the following could address your concerns.
>
> > **The method is currently only tested in relatively simple MPE environments**
>
> MPE environments emphasize different interaction types: competitive, cooperative, and mixed, fostering diverse agent dynamics. In future work, we plan to apply our methods to other environments, such as StarCraft. In terms of agent number, we already conduct  a task with 8 agents in a 4x4 grassland setting. Additionally, we extended our analysis to a 10x10 grassland test, with the results presented below.
>
> > **As the number of agents increases, the computational complexity of inferring attention weights for multiple agents may become a bottleneck. The paper does not fully address how the method scales with a larger number of agents or more diverse agent types.**
>
> The inverse-att network will scale up in the benefit of the attention structure. We present the results for 20 agents in the Grassland environment with a 10-10 scale. The Inverse-Att agents not only scale effectively but also outperform other ablations and baseline methods.
>
>
> |                     | MAPPO   | IPPO    | MAA2C   | ToM2C   | Self-Att | Inverse-Att |
> |---------------------|---------|---------|---------|---------|----------|-------------|
> | **Grassland Sheep** | -68.15  | -74.64  | -55.00  | -58.96  | -143.50  | -52.34      |
> | **Grassland Wolf**  | 90.47   | 49.81   | 30.50   | 30.21   | 120.08   | 132.39      |
>
>
> > **The method assumes that agents are of the same type, which limits its generalization.**
>
> We do not assume that all agents are of the same type. For instance, in a competitive environment like Adversary, there are two distinct types of agents: sheep and wolves. Since inverse attention employs a goal-focused (GF) representation, this mechanism can be applied to all types of teammates, regardless of their roles.
>
> > **The inverse attention mechanism is trained in an offline manner, which may limit its adaptability in a rapidly changing environment.**
>
> Inverse attention is initially trained offline to abstract the relationships between agents' observations and actions. However, an online training phase is introduced later (as described in the second while loop of Algorithm 1), where MARL is applied to adapt more effectively to other agents and dynamic environments. Experimental results in Tables 1–5 demonstrate that, even when interacting with various unseen agents or human players, inverse attention consistently outperforms all other baseline methods.
>
> > **Does the performance of inverse attention agents degrade as the number of agents increases, particularly in larger environments like the 4-4 Grassland game?**
>
> The 4-4 and 10-10 results show that inverse-att agents outperform the other baselines with a large margin which could scale up as the agent number increases.
>
> > **Why does the cooperative gain between multiple inverse attention agents not increase linearly?**
>
> Yes, we believe cognitive loops or mutual inference inaccuracies indeed plays a role in the absence of linear reward increase.
>
> > **Why do inverse attention agents perform inconsistently when cooperating with humans in some roles (e.g., sheep in Adversary), and how can this cooperation be made more robust?**
>
> We did not observe any inconsistencies in the behavior of Inverse-Att agents, including the sheep in the Adversary environment. When comparing them to other methods, Inverse-Att agents consistently achieve much higher rewards with relatively low variance compared to all other baselines. Could you clarify what you mean by "not robust"?
>
> > **How is the accuracy of the inverse attention network related to task success, and why doesn't high inference accuracy always correlate with better performance in tasks like Adversary?**
>
> The higher accuracy of the inverse attention network allows agents to achieve better performance by reducing the noise in their input. This advantage applies across all tasks, including competitive scenarios like Adversary. For instance, if Agent A knows that teammates B and C are moving left to avoid a wolf, it might decide to go right, avoiding the crowd and reducing its risk of being caught.

---

> ### Author Response · Authors · 2024-11-23
> **Response to Reviewer HVvH Continued**
>
> > **Performance Difference Between Inverse Attention Agents and Baseline Agents**
>
> At the network structure level, Inverse Attention Agents utilize the attention mechanism, while baseline methods (e.g., MAPPO, IPPO, MAA2C, ToM2C) rely on MLPs. MLPs are input-sensitive, learning fixed mappings between specific input configurations and outputs, which makes them less effective in scenarios with dynamic teammates or opponents. In contrast, attention mechanisms are input-invariant, dynamically computing relevance across entities without depending on fixed mappings.
> At the methodological level, the Inverse Attention approach models agents by inferring their goals. By integrating this inferred information with environmental data, it constructs a more comprehensive state representation, leading to improved action decisions.
>
> > **Inverse Attention Performance in Competitive Tasks and Cooperative Tasks**
>
> We have no evidence showing that the inverse attention mechanism performs better in competitive tasks than in cooperative ones. Our experiment results show that Inverse Attention Agents outperform baselines by a margin in every cross competitions, demonstrating its superior performance in both competitive and cooperative tasks.

---

> > ### Author Response · Authors · 2024-11-24
> > **Follow-up the discussion**
> >
> > Dear Reviewer HVvH,
> >
> > We sincerely thank you for taking the time to review our work and for providing valuable feedback. We are especially grateful for your recognition of our efforts, as reflected in your updated score. Your constructive insights have been instrumental in helping us refine our work further.
> >
> > Please feel free to let us know if there are any additional suggestions or areas you feel we could improve. We deeply value your input.
> >
> > Best regards,
> >
> > The Authors

---

### Official Review · Reviewer_MQhb · 2024-11-04

**Soundness:** 3
**Presentation:** 3
**Contribution:** 2
**Rating:** 3
**Confidence:** 4

**Summary:**

This paper studies the setting similar to ad hoc teamwork, i.e., the trained agents encounter unfamiliar agents when performing tasks. It proposes an inverse attention network that infers the attention of other agents on each goal based on their observations and prior actions. Doing so helps maintain consistency in decision-making across different scenarios and the agent can adjust its final action by refining the attention weights given the outputs of the inverse attention network.

**Strengths:**

1. This work uses the gradient field (GF) representations to represent the goals of the agent within specific environments, which is an interesting idea.

2. This work conducts a human study to demonstrate the effectiveness of the proposed method.

**Weaknesses:**

1. The proposed inverse attention network requires the observation of agent $j$ when inferring the attention weights of agent $j$. Therefore, the proposed method only works in fully observable environments and will face problems in partially observable environments if the observations of other agents are unavailable.

2. The proposed method assumes the observations of an agent can be decomposed into a combination of $N$ goals within the environment. This assumption holds for the tested environments in MPE. However, in many environments, there are no explicit multiple goals, and how to decompose an observation into $N$ goals is unclear. Therefore, the proposed method has limited applications.

3. When training the inverse attention network, this work only uses the agent $i$’s own data. This will introduce bias when the agent $i$'s experience cannot cover the trajectories of other agents.

4. The MPE environments are very simple and similar to each other. The scalability test only tries at most 4 agents which is small.

5. This work does not compare with the existing works form the ad hoc teamwork domain.

**Questions:**

Please refer to the above weakness section.

---

> ### Author Response · Authors · 2024-11-23
> **Response to Reviewer MQhb**
>
> We thank you for your insightful comments on improving our paper. We hope the following could address your concerns.
>
> > **The proposed inverse attention network requires the observation of agent $j$ when inferring the attention weights of agent $j$**
>
> This approach can be extended to partial observations by applying inverse attention only to agents within the observable range. Since the inverse attention network leverages the attention mechanism, it is more robust to a dynamic number of inputs. We demonstrate the impact of partial observation by testing different visibility radii in all tested games (spread, adversary, grassland, navigation, tag): 1.5, 1, and 0.5. Across all three settings, agents using inverse attention consistently achieve the best performance.
>
> ### Radius=1.5
>
> |                     | MAPPO   | IPPO    | MAA2C   | ToM2C   | Self-Att | Inverse-Att |
> |---------------------|---------|---------|---------|---------|----------|-------------|
> | **Spread**          | 35.95   | 0.50    | 48.90   | 36.96   | 178.03   | 263.53      |
> | **Adversary Sheep** | -62.63  | -72.94  | -66.36  | -59.14  | -19.72   | -17.25      |
> | **Adversary Wolf**  | 40.83   | 13.83   | 53.56   | 40.92   | 57.64    | 94.18       |
> | **Grassland Sheep** | -54.99  | -73.85  | -58.71  | -59.48  | -12.24   | 15.91       |
> | **Grassland Wolf**  | 35.94   | 18.87   | 64.82   | 27.78   | 69.40    | 97.21       |
> | **Navigation**      | 317.39  | 222.81  | 307.55  | 300.87  | 409.54   | 467.17      |
> | **Tag Sheep**       | -58.12  | -100.08 | -56.37  | -50.72  | -13.18   | -6.93       |
> | **Tag Wolf**        | 38.66   | 18.25   | 49.21   | 45.46   | 66.83    | 70.50       |
>
> ### Radius=1
>
> |                     | MAPPO   | IPPO    | MAA2C   | ToM2C   | Self-Att | Inverse-Att |
> |---------------------|---------|---------|---------|---------|----------|-------------|
> | **Spread**          | 44.90   | 0.66    | 53.72   | 40.47   | 78.22    | 111.40      |
> | **Adversary Sheep** | -37.92  | -58.72  | -60.51  | -46.10  | -21.97   | -20.27      |
> | **Adversary Wolf**  | 42.79   | 14.01   | 39.68   | 32.66   | 39.62    | 77.37       |
> | **Grassland Sheep** | -45.45  | -66.14  | -57.12  | -54.01  | -2.26    | -20.41      |
> | **Grassland Wolf**  | 38.60   | 19.70   | 67.20   | 29.24   | 69.23    | 74.85       |
> | **Navigation**      | 267.76  | 162.79  | 262.10  | 237.39  | 243.65   | 269.83      |
> | **Tag Sheep**       | -42.62  | -105.53 | -50.67  | -51.19  | -16.01   | -8.13       |
> | **Tag Wolf**        | 38.63   | 16.59   | 37.14   | 40.14   | 64.57    | 79.42       |
>
>
> ### Radius=0.5
>
> |                     | MAPPO   | IPPO    | MAA2C   | ToM2C   | Self-Att | Inverse-Att |
> |---------------------|---------|---------|---------|---------|----------|-------------|
> | **Spread**          | 53.28   | 0.60    | 22.73   | 47.36   | 19.22    | 59.16       |
>
> |                     | MAPPO   | IPPO    | MAA2C   | ToM2C   | Self-Att | Inverse-Att |
> |---------------------|---------|---------|---------|---------|----------|-------------|
> | **Adversary Sheep** | -16.28  | -39.25  | -52.27  | -26.90  | -19.50   | -13.47      |
> | **Adversary Wolf**  | 34.96   | 11.86   | 25.16   | 19.73   | 23.90    | 51.27       |
> | **Grassland Sheep** | -34.30  | -55.67  | -53.54  | -39.50  | -23.32   | -11.73      |
> | **Grassland Wolf**  | 40.87   | 21.57   | 50.94   | 28.99   | 57.40    | 61.80       |
> | **Navigation**      | 163.95  | 146.04  | 162.58  | 226.77  | 253.06   | 256.57      |
> | **Tag Sheep**       | -20.69  | -93.58  | -36.80  | -53.28  | -14.03   | -6.86       |
> | **Tag Wolf**        | 30.28   | 15.57   | 54.98   | 29.60   | 39.51    | 58.07       |
>
>
> > **When training the inverse attention network, this work only uses the agent i’s own data.**
>
> This setting is more challenging and realistic, as in most real-world environments like card games or hunting, it is impossible to access the states of other agents. Agents can only rely on their own observations, making it equally unfeasible to determine the exact goals of other agents. In this context, inverse attention provides a viable approach to modeling other agents through their inferred goals, offering a more generalized method for representing their behaviors.
>
> > **This work does not compare with the existing works from the ad hoc teamwork domain.**
>
> The Tom2C agent is from the adhoc domain, which belongs to the “agent modeling” category according to https://arxiv.org/abs/2202.10450.

---

> ### Author Response · Authors · 2024-11-23
> **Response to Reviewer MQhb Continued**
>
> > **The MPE environments are very simple and similar to each other. The scalability test only tries at most 4 agents which is small.**
>
> MPE environments emphasize different interaction types: competitive, cooperative, and mixed, fostering diverse agent dynamics. In future work, we plan to apply our methods to other environments, such as StarCraft. In terms of agent number, we already conduct  a task with 8 agents in a 4x4 grassland setting. Additionally, we extended our analysis to a 10x10 grassland test.
>
> |                     | MAPPO   | IPPO    | MAA2C   | ToM2C   | Self-Att | Inverse-Att |
> |---------------------|---------|---------|---------|---------|----------|-------------|
> | **Grassland Sheep** | -68.15  | -74.64  | -55.00  | -58.96  | -143.50  | -52.34      |
> | **Grassland Wolf**  | 90.47   | 49.81   | 30.50   | 30.21   | 120.08   | 132.39      |
>
> > **In many environments, there are no explicit multiple goals, and how to decompose an observation into $N$ goals is unclear.**
>
> Goals are not needed. We follow [1] to use gradient fields, and we treat Goal-Focused Features (GFs) as proxies for goals.
>
> [1] SocialGFs: Learning Social Gradient Fields for Multi-Agent Reinforcement Learning

---

> > ### Author Response · Authors · 2024-11-24
> > **Follow-up the discussion**
> >
> > Dear Reviewer MQhb,
> >
> > We greatly appreciate your time and feedback on our work. We have carefully addressed your comments and clarified potential misunderstandings. Additionally, we also included new experimental results to corroborate our findings.
> >
> > We kindly invite you to revisit our paper in light of these updates and clarifications. We would greatly appreciate it if you could consider whether our responses warrant a reevaluation of your rating.
> >
> > Best regards,
> >
> > The Authors

---

> > > ### Comment · Reviewer_MQhb · 2024-11-25
> > >
> > > I thank the authors for the rebuttal. Regarding the use of agent $j$'s observation for computing attention weights and the use of agent $i$'s own data for training, I think this is the inherent design of this work and it is hard to observe their adverse effects in simple environments like MPE. Also, I would not consider Tom2C as a canonical work for ad hoc teamwork because ad hoc teamwork does not use explicit communications. There are many existing ad hoc teamwork works that can be compared with. In the end, regarding the goal of the environment,  there are texts in line 240: "... observation of agent i, which can be decomposed into a combination of N goals" and line 339: "resulting in a gf representation structured as {gf1, gf2, ..., gfN , gfwall}". How to decompose the observation into $N$ parts and how to determine the $N$? In experiments, $N$ directly corresponds to the number of entities, e.g., landmark, sheep, and wolves.

---

### Official Review · Reviewer_KzQg · 2024-11-04

**Soundness:** 3
**Presentation:** 3
**Contribution:** 2
**Rating:** 5
**Confidence:** 4

**Summary:**

This paper proposes a method for developing training a parameterized function to predict the agents attention weights in a goal oriented task proposed in Long et al 2020. The paper proposes to do that using an inverse attention network. A dataset is created from the most recently accumulated attention weights with respect to their local observations, and this is then used to train the inverse attention network to predict agent attention weights during inference. Additionally, the authors show that the inverse attention network, due to its simplicity, has a high prediction accuracy after convergence.

The agents have access to the local observation of each of the other agents, which it uses to predict the intentions of other agents, and alter its own actions accordingly. Experimental results show the efficacy of the inverse attention mechanism when compared to standard non communication based algorithms.

**Strengths:**

The proposed method approaches the problem of inferring intentions of agents by allowing each agent to access the observation of other agents and reason about what their intentions are. The approach offers a simple yet effective tool for multi-agent systems especially for heavily correlated environments, for example in dense environments. Altering agent behavior through explicit reasoning over other agents intentions is of great interest to the multi-agent community.

**Weaknesses:**

As part of the scalability analysis, a maximum number of 8 agents have been tested. Due to the concatenation of each agents attention weight, I would like an analysis on the limitations of the scalability, particularly for the accuracy of IW predictions and the complexity of processing each pairwise agent inference separately as number of agents grow to 20~50 agents.

**Questions:**

The paper claims that agents generally perform poorly when unseen states are communicated by agents (heterogeneous or otherwise). The paper focuses on predicting just the attention weights instead of the actual hidden states to mitigate the above issue. Am I correct in assuming that as part of the IW networks training, each independent agent should have had encountered those states for it to make an accurate prediction of the other agents attention weights.

---

> ### Author Response · Authors · 2024-11-23
> **Response to Reviewer KzQg**
>
> We thank you for your insightful comments on improving our paper. We hope the following could address your concerns.
>
> > **Scalability Analysis**
>
> We present the results for 20 agents in the Grassland environment with a 10-10 scale. The Inverse-Att agents not only scale effectively but also outperform other ablations and baseline methods.
>
> |                     | MAPPO   | IPPO    | MAA2C   | ToM2C   | Self-Att | Inverse-Att |
> |---------------------|---------|---------|---------|---------|----------|-------------|
> | **Grassland Sheep** | -68.15  | -74.64  | -55.00  | -58.96  | -143.50  | -52.34      |
> | **Grassland Wolf**  | 90.47   | 49.81   | 30.50   | 30.21   | 120.08   | 132.39      |
>
> > **Am I correct in assuming that as part of the IW networks training, each independent agent should have encountered those states for it to make an accurate prediction of the other agent's attention weights.**
>
> This is an excellent question, as current methods do not explicitly model other agents. Instead, we rely on inverse attention mechanisms to represent other agents. In this context, the "state" refers to a higher-level abstraction that encompasses both "other agents" and their representations, which are framed as inverted goals. This approach has the potential to generalize to unseen states of previously unobserved agents, much like how multi-agent reinforcement learning (MARL) can, to some extent, generalize to unseen physical states.

---

> > ### Author Response · Authors · 2024-11-24
> > **Follow-up the discussion**
> >
> > Dear Reviewer KzQg,
> >
> > We greatly appreciate your time and feedback on our work. We have carefully addressed your comments and clarified potential misunderstandings. Additionally, we also included new experimental results to corroborate our findings.
> >
> > We kindly invite you to revisit our paper in light of these updates and clarifications. We would greatly appreciate it if you could consider whether our responses warrant a reevaluation of your rating.
> >
> > Best regards,
> >
> > The Authors

---

### Official Review · Reviewer_kZyt · 2024-11-04

**Soundness:** 2
**Presentation:** 3
**Contribution:** 2
**Rating:** 5
**Confidence:** 2

**Summary:**

The paper presents an end-to-end method based on the attention mechanism to enable the easy adaptation of trained agents to different environments. The authors provided evaluations of model interactions with artificial and human agents.

**Strengths:**

- The paper proposes a novel approach to attention mechanism in the multi-agent system setting.
- The results are tested both with unseen artificial agents and with human agents' interaction with the proposed agent.

**Weaknesses:**

- it is not clear to me and was not discussed in the text how the agent observation can be decomposed into a combination of goals.
- The authors collect the attention inference dataset but do not provide an analysis of how different realizations of such attention weights collections may influence the resulting inverse-attention agent performance (e.g. dataset collected with models with different training hyperparameters, random seeds, attention sizes).
- The authors provide no ablation study.

**Questions:**

- The reference of the "Attention is all you need" paper has an incorrect year of publication.
- The font sizes on figures 2 and 3 should be increased to improve readability.

---

> ### Author Response · Authors · 2024-11-23
> **Response to Reviewer kZyt**
>
> We thank you for your insightful comments on improving our paper. We hope the following could address your concerns.
>
> > **It is not clear to me and was not discussed in the text how the agent observation can be decomposed into a combination of goals.**
>
> This is discussed in Section 3.2, and in Supplementary A.3, where we provide a detailed example of how to construct the dataset for goal-focused (GF) training. The GFs are treated as goals, allowing agents to directly act based on these goals.
>
> > **The authors collect the attention inference dataset but do not provide an analysis of how different realizations of such attention weights collections may influence the resulting inverse-attention agent performance**
>
> We collect attention inference dataset from part of the Self_Att agent's training trajectory. All Self_Att agents are trained with the same hyperparameters, and we believe the current training trajectory is sufficient to cover majority cases, so we do not believe that changing random seeds would be necessary. Additionally, the number of agents is fixed, so we do not need to train inverse attention networks with different number of attention weights.
>
> > **The authors provide no ablation study.**
>
> Inverse-Att features two novel modules: the gradient field and the inverse attention network. Self-Att is an ablation of the inverse attention network, included as a baseline in the main paper. As shown in experiment sections, it consistently outperforms all other baseline models.
>
> The ablation study for Inverse-Att without the gradient field is extensively discussed in [1]. Additionally, we conducted experiments in the Spread 3-agents game, Adversary $3$-$3$ game, and Grassland $3$-$3$ game, replacing the gradient field with relative positions for Inverse-Att agents only. This variant is referred to as Inverse-Att-Vanilla.
>
> In our evaluation, Inverse-Att-Vanilla competed against the same set of baseline agents: MAPPO, IPPO, MAA2C, ToM2C, and Self-Att. For Inverse-Att-Vanilla agents, the input consisted solely of raw environmental observations, while all other agents retained gradient field (GF) inputs. The results of these experiments are presented in the updated Appendix, which details the performance of models competing with Inverse-Att-Vanilla. Furthermore, Table 13 in the Appendix compares the performance of Inverse-Att-Vanilla and Inverse-Att against the same set of agents in identical agent pools. The findings demonstrate that while Inverse-Att delivers strong performance compared to other baselines, it outperforms all other agents in most cases. This highlights the critical importance of incorporating GF representations into the observation space.
>
> **Ablation for GFs**
> |                | MAPPO  | IPPO   | MAA2C  | ToM2C  | Self-Att | Inverse-Att-Vanilla |
> |----------------|--------|--------|--------|--------|----------|---------------------|
> | **Spread**     | 27.24  | 0.65   | 43.21  | 27.38  | 254.04   | 348.66              |
> | **Adversary Sheep** | -101.54 | -118.97 | -106.39 | -104.60 | -24.56   | -19.42              |
> | **Adversary Wolf**  | 34.91  | 16.99  | 90.17  | 51.44  | 90.86    | 95.17               |
> | **Grassland Sheep** | -76.60 | -92.61 | -73.03 | -81.51 | -1.01    | 21.50               |
> | **Grassland Wolf** | 24.23  | 14.85  | 79.99  | 24.46  | 100.66   | 103.05              |
>
> **Comparison between Inverse-Att-Vanila and Inverse-Att agents**
> |                     | Inverse-Att-Vanilla | Inverse-Att |
> |---------------------|---------------------|-------------|
> | **Spread**          | 348.66             | 404.14      |
> | **Adversary Sheep** | -19.42             | -16.91      |
> | **Adversary Wolf**  | 95.17              | 110.15      |
> | **Grassland Sheep** | 21.50              | 28.59       |
> | **Grassland Wolf**  | 103.05             | 101.21      |
>
> > **The reference of the "Attention is all you need" paper has an incorrect year of publication.**
>
> Thank you for pointing out this oversight. We have revised our paper to correctly cite it.
>
> > **The font sizes on figures 2 and 3 should be increased to improve readability**
>
> Thank you for your suggestion, we have increased the font size and updated the figures in the paper.
>
>
>
>
> [1] SocialGFs: Learning Social Gradient Fields for Multi-Agent Reinforcement Learning

---

> > ### Author Response · Authors · 2024-11-24
> > **Follow-up the discussion**
> >
> > Dear Reviewer kZyt,
> >
> > We greatly appreciate your time and feedback on our work. We have carefully addressed your comments and clarified potential misunderstandings. Additionally, we also included new experimental results to corroborate our findings.
> >
> > We kindly invite you to revisit our paper in light of these updates and clarifications. We would greatly appreciate it if you could consider whether our responses warrant a reevaluation of your rating.
> >
> > Best regards,
> >
> > The Authors

---

> > > ### Comment · Reviewer_kZyt · 2024-12-02
> > >
> > > Dear Authors,
> > >
> > > Thank you for your clarifications. My questions were fully addressed and I have adjusted my scores accordingly.

---

### Meta-Review · Area_Chair_ZrfF · 2024-12-19

**Metareview:**

This paper presents the problem of estimating the attention of other agents in a multi-agent setting. The proposed significance is that estimating attention of other agents is proposed to be a bottleneck to operating with unfamiliar intentions. The method is tested on ad-hoc collaboration tasks, aiming to show that it outperforms other ad-hoc collaboration methods. Experiments in a variety of 'Multi-agent Particles Environments' (MPE) show that the method performs substantially better than the included methods for comparison.

### Strengths
Reviewers commented that the approach is novel, the results are tested with both unseen artificial and human agents, that the approach is simple and effective, altering agent behavior by explicitly reasoning over other agents' intentions is of great interested to the multi-agent community, that the mathematical representation of goals is interesting, that the method is well-justified and experimental setup thorough, the paper is clearly written, and the ability to improve agent cooperation in ad-hoc settings is important.

### Weaknesses
Reviewers commented that it was unclear how the agent observation can be decomposed into a combination of goals, missing comparisons to different realizations of attention weights, missing ablation study, a missing analysis on the scalability of the method, that the method requires the observation of agent $j$ when inferring the attention weights of agent $j$ (from further discussion this reviewer pointed out that this method requires the assumption that the agent knows the count $N$ of other agents and how to decompose it into $N$ parts), and the method is tested only in relatively simple environments with small numbers of agents.

I have evaluated the concerns of the reviewers in light of the reviewer discussion and describe them below. The authors provided fairly complete and substantial responses to every concern, in my opinion, yet there was limited engagement with the author responses. After evaluation, this is a tough borderline case in which the average score from the reviewers is somewhat at odds with the results of the reviewer discussion. Given that the paper makes fairly clear claims that the reviewers agreed were interesting, the reviewers thought that the approach was novel and of interest to the community, and that the evidence supports the claims (although was a bit limited to somewhat smaller experimental environments), I recommend to err on the side of acceptance.

**Additional Comments On Reviewer Discussion:**

Reviewer kZyt (final rating 5) commented:
> Thank you for your clarifications. My questions were fully addressed and I have adjusted my scores accordingly.

and updated their rating to a 5, without justifying what further concerns or questions remained. Therefore, I take their evaluation as a slightly positive evaluation.

Reviewer KzQg (final rating 5) did not engage with author responses, and their only stated weakness was the absence of results and a complexity analysis with 20-50 agents. The authors supplied an experiment with 20 agents, although did not supply the computational complexity of the overall approach, which would strengthen the paper. Thus, I consider this reviewer's concerns mostly resolved.

Reviewer MQhb (final rating 3) was looking for clarifications about the assumptions required to partition the observation, comparisons to more agents than 4 and more complex settings, and also was concerned about lacking some comparisons to other ad hoc teamwork approaches. The authors also pointed out comparisons to 8 agents in the existing work, and extended it to a 10x10 grassland test. To the final issue, the authors responded that the ToM2C approach they compared against is one such ad hoc teamwork approach, and the reviewer did not respond with any other explicit requests for additional comparisons. Therefore I consider the reviewer's comparison request adequately resolved. The authors did not respond to a late-breaking comment from the reviewer concerning the assumptions (although the authors had 2 days to do so). It's not clear to me that these assumptions are alone an obvious reason for rejection. Thus, I consider most, but not all, of this reviewer's concerns resolved. The reviewer, however, did not update their final rating.

Reviewer HVvH (final rating 6) was mainly looking for more extensive and complex evaluations. The authors responded by explaining why the existing evaluation is nontrivial, and addressed concerns regarding the complexity. The reviewer did not engage with the author responses.

---

### Decision · Program_Chairs · 2025-01-22

Accept (Poster)